# Understanding the Failure of Batch Normalization for Transformers in NLP

**Jiaxi Wang**[1], **Ji Wu**[1,2], **Lei Huang**[3]

[1]Department of Electronic Engineering, Tsinghua University
[2]Institute for Precision Medicine, Tsinghua University
{wjx20@mails, wuji_ee@mail}.tsinghua.edu.cn
[3]SKLSDE, Institute of Artificial Intelligence, Beihang University
huangleiAI@buaa.edu.cn

## Abstract

Batch Normalization (BN) is a core and prevalent technique in accelerating the training of deep neural networks and improving the generalization on Computer Vision (CV) tasks. However, it fails to defend its position in Natural Language Processing (NLP), which is dominated by Layer Normalization (LN). In this paper, we are trying to answer why BN usually performs worse than LN in NLP tasks with Transformer models. We find that the inconsistency between training and inference of BN is the leading cause that results in the failure of BN in NLP. We define Training Inference Discrepancy (TID) to quantitatively measure this inconsistency and reveal that TID can indicate BN's performance, supported by extensive experiments, including image classification, neural machine translation, language modeling, sequence labeling, and text classification tasks. We find that BN can obtain much better test performance than LN when TID keeps small through training. To suppress the explosion of TID, we propose Regularized BN (RBN) that adds a simple regularization term to narrow the gap between batch statistics and population statistics of BN. RBN improves the performance of BN consistently and outperforms or is on par with LN on 17 out of 20 settings, involving ten datasets and two common variants of Transformer[1].

## 1 Introduction

Deep learning [19] has revolutionized Computer Vision (CV) [18] and Natural Language Processing (NLP) [39]. Normalization layers are key components to stabilize and accelerate the training in Deep Neural Networks (DNNs). In CV, Batch Normalization (BN) [15] is the default normalization technique and reveals superior performance over other normalization techniques in image recognition tasks by enforcing the input of a neuron to have zero mean and unit variance within a mini-batch data. Furthermore, a growing number of theoretical works analyze the excellent properties of BN in benefiting optimization [15, 34, 4, 12, 7, 8]. While BN almost dominates in CV with empirical success and theoretical properties, Layer Normalization (LN) is the leading normalization technique in NLP, especially for Transformer models that achieve the state-of-the-art performance on extensive tasks, including machine translation [39], natural language understanding [9], text generation [32], few shot learning [5], to name a few. As a direct substitute of LN, BN performs poorly in Transformer for neural machine translation [36]. It remains elusive to explain the failure of BN in NLP community. In this work, we are trying to take a step forward. Our contributions are summarized as follows:

---

[1]Our code is available at `https://github.com/wjxts/RegularizedBN`

- We find that the inconsistency between training and inference leads to the failure of BN in NLP, supported by our extensive experiments, including image classification, neural machine translation, language modeling, sequence labeling, and text classification tasks.
- We define Training Inference Discrepancy (TID) to quantitatively measure this inconsistency and show that TID can serve as an indicator of BN's performance. In particular, BN reaches much better test performance than LN when TID keeps small through training, *e.g.*, in image recognition and language modeling tasks.
- We propose Regularized BN (RBN) that adds a regularization term in BN to penalize and reduce the TID when the TID of BN is large. We reveal the optimization advantages of RBN over LN by exploring the layer-wise training dynamics of Transformer.
- We empirically show that RBN can exceed or match the performance of LN, sometimes with a large margin, on 17 out of 20 settings, involving ten datasets and two common variants of Transformer. Besides, RBN introduces no extra computation at inference compared to LN.

## 2 Related Work

**Analyses of BN's Success**    As BN becomes an indispensable component in deep neural networks deployed in CV tasks, a bunch of works explore the theoretical reasons behind its success. From the view of optimization, the original BN paper [15] argues that BN can reduce internal covariate shift and thus stabilize the training, while Santurkar et al. [34] debate that BN could smooth the loss landscape and thus enable training of neural network with larger learning rate [4]. Daneshmand et al. [7, 8] prove that a stack of randomized linear layers and BN layers will endow the intermediate features of neural network with sufficient numerical rank as depth increases, which is beneficial for optimization and learning discriminative hierarchical features. Huang et al. [12] show that BN could improve the layer-wise conditioning of the neural network optimization by exploring the spectrum of Hessian matrix with block diagonal approximation [26]. From the view of generalization, Ioffe and Szegedy [15], Luo et al. [23], Li et al. [20], Wu and Johnson [41] argue that BN serves as regularizer which reduces over-fitting when its stochasticity is small and may have detrimental effect when it is large [41]. Huang et al. [11] further propose Stochastic Normalization Disturbance (SND) to measure such stochasticity and shows that large SND will hinder the training of neural networks.

**Training Inference Inconsistency of BN**    Normalizing along the batch dimension usually introduces training inference inconsistency since mini-batch data is neither necessary nor desirable during inference. BN uses population statistics, estimated by running average over mini-batch statistics, for inference. The training inference inconsistency usually harms the performance of BN for small-batch-size training since the estimation of population statistics could be inaccurate [40]. One way to reduce the inconsistency between training and inference is to exploit the estimated population statistics for normalization during training [14, 6, 45, 48, 47]. These works may outperform BN when the batch size is small, where inaccurate estimation may be the main issue [15, 16], but they usually work inferior to BN under moderate batch-size training [22]. Another way to reduce the inconsistency is estimating corrected normalization statistics during inference only, either for domain adaptation [21], corruption robustness [35, 29, 2], or small-batch-size training [37, 38]. We note that a recent work [13] investigates the estimation shift problem of BN. Unlike this work that addresses the accumulated estimation shift due to the stack of BNs for CNNs in CV tasks, our work pays more attention to how the training inference inconsistency of BN correlates with its performances for Transformers in NLP tasks. Besides, the estimation shift of BN defined in [13], which addresses the differences between the estimated population statistics and the expected statistics, differs from our TID of BN that addresses the differences between the mini-batch statistics and populations statistics.

**Exploring the Failure of BN in Transformer**    Similar to our work, Power Normalization (PN) [36] also investigates the reason behind the failure of BN in Transformers. Our work significantly differs from PN [36] in the following facets. PN attributes the failure of BN to the unstable training of BN incurred by fluctuated forward and backward batch statistics with outlier values, while we observe that the training of BN is as good as LN and the inconsistency between training and inference of BN matters more. Based on our observation, we propose a regularization term to reduce the TID of BN. Compared with PN, which incorporates a layer-scale layer (root mean square layer normalization [49] without affine transformation [43]), our method introduces no extra computation at inference. Besides, we use a more reasonable index to measure inconsistency which is invariant to the scale of data. Furthermore, we show that our RBN can improve the layer-wise training dynamics of LN, which reveals the optimization advantages of RBN.

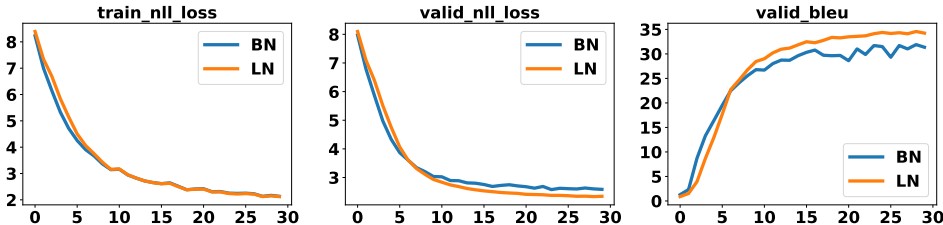

Figure 1: Train loss, validation loss/BLEU of Transformer trained on IWSLT14 with BN and LN. The training of Transformer$_{BN}$ is better than Transformer$_{LN}$ while the validation loss/BLEU of Transformer$_{BN}$ underperforms that of Transformer$_{LN}$ after 8 epoch. At the end of the training, Transformer$_{BN}$ falls behind Transformer$_{LN}$ with large BLEU scores. Lower loss and higher BLEU scores indicate better performance. Based on the inconsistency of training and validation performance of BN, we hypothesize that the training inference discrepancy of BN causes its performance degradation.

## 3 Analyses of Training Inference Inconsistency in Transformer$_{BN}$

### 3.1 Preliminary

Batch Normalization (BN) [15] is typically used to stabilize and accelerate DNN's training. Let $\mathbf{x} \in \mathbb{R}^d$ denote the $d$-dimensional input to a neural network layer. During training, BN standardizes each neuron/channel within $m$ mini-batch data by[2]

$$\hat{\mathbf{x}}_j = BN_{train}(\mathbf{x}_j) = \frac{\mathbf{x}_j - \mu_{B,j}}{\sqrt{\sigma_{B,j}^2}}, \;\; j = 1, 2, ..., d, \tag{1}$$

where $\mu_{B,j} = \frac{1}{m}\sum_{i=1}^{m}\mathbf{x}_j^{(i)}$ and $\sigma_{B,j}^2 = \frac{1}{m}\sum_{i=1}^{m}(\mathbf{x}_j^{(i)} - \mu_{B,j})^2$ are the mini-batch mean and variance for each neuron, respectively. Note that an extra small number $\epsilon$ is usually added to the variance in practice to prevent numerical instability. During inference, the population mean $\mu$ and variance $\sigma^2$ of the layer input are required for BN to make a deterministic prediction [15] as:

$$\hat{\mathbf{x}}_j = BN_{inf}(\mathbf{x}_j) = \frac{\mathbf{x}_j - \mu_j}{\sqrt{\sigma_j^2}}, \;\; j = 1, 2, ..., d. \tag{2}$$

These population statistics $\{\mu, \sigma^2\}$ are usually calculated as the running average of mini-batch statistics over different training iteration $t$ with an update factor $\alpha$ as follows:

$$\begin{cases} \mu^{(t)} = (1-\alpha)\mu^{(t-1)} + \alpha\mu_B^{(t)}, \\ (\sigma^2)^{(t)} = (1-\alpha)(\sigma^2)^{(t-1)} + \alpha(\sigma_B^2)^{(t)}. \end{cases} \tag{3}$$

The discrepancy of BN for normalization during training (using Eqn. 1) and inference (using Eqn. 2) can produce stochasticity, since the population statistics of BN are estimated from the mini-batch statistics that depend on the sampled mini-batch inputs. This discrepancy is believed to benefit the generalization [15, 11] if the stochasticity is well controlled. However, this discrepancy usually harms the performance of small-batch-size training [40] since the estimation of population statistics can be inaccurate. To address this problem, a bunch of batch-free normalizations are proposed that use consistent operations during training and inference, *e.g.*, Layer Normalization (LN) [1].

**Basic Observations** To analyze the failure of BN in NLP tasks, we first plot the training loss and validation loss/BLEU [31] of BN and LN on IWSLT14 (De-En) dataset with the original Transformer model (see Figure 1). We observe that the training of Transformer$_{BN}$ is faster than Transformer$_{LN}$. The training nll_loss of BN is even smaller than that of LN, especially at the beginning. However, validation loss/BLEU of BN is worse than that of LN after around the seventh epoch. This phenomenon can not be attributed to over-fitting since BN introduces more stochasticity than LN in the training phase. The inconsistency between training and inference of BN may play a role.

---

[2]BN usually uses extra learnable scale and shift parameters [15] to recover the potentially reduced representation capacity, and we omit them since they are not relevant to our discussion.

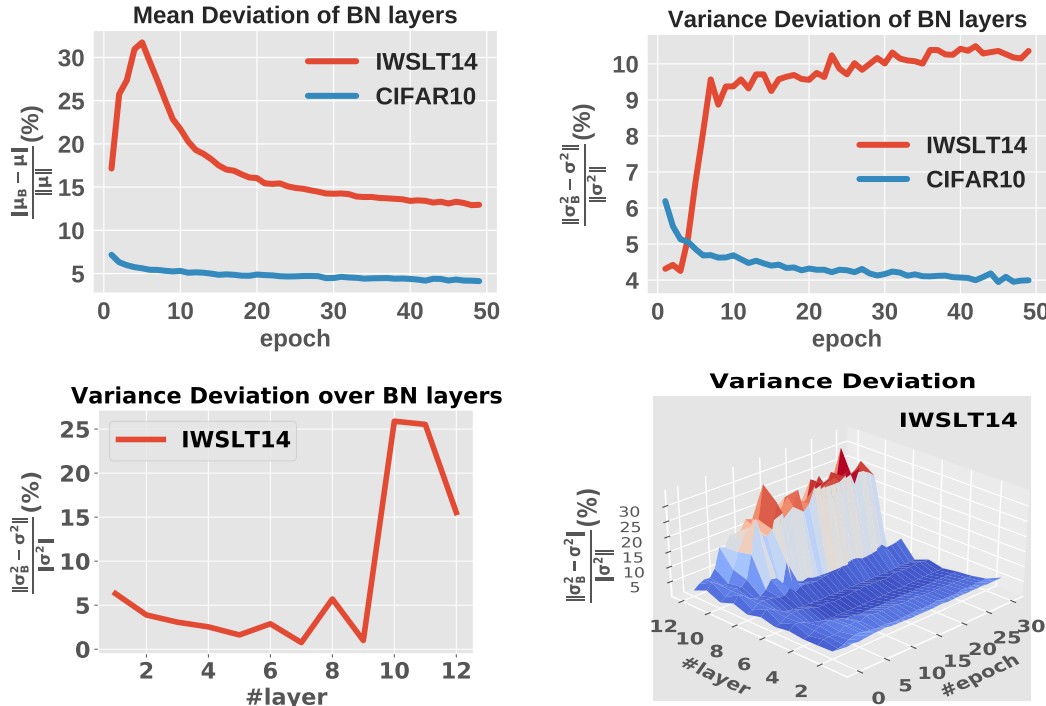

Figure 2: Top: The average deviation of batch mean $\mu_B$ (left figure) and batch variance $\sigma_B^2$ (right figure) to population mean $\mu$ and population variance $\sigma^2$ of all BN layers through training in ResNet18 and Transformer$_{BN}$. There are 21 BN layers in ResNet18 and 12 BN layers in the encoder of Transformer$_{BN}$. At the end of training, ResNet18 has mean/variance deviation of around $4\%/4\%$ and those in Transformer$_{BN}$ are around $11\%/13\%$. Large deviation of statistics hurts the performance of Transformer$_{BN}$. Bottom: Variance deviation of BN layers with different depths (left) at the end of training and variance deviation over depth and training progress (right).

Since BN in ResNet18 also involves training inference inconsistency, we guess the degree of such inconsistency has a difference between ResNet18 and Transformer$_{BN}$. Therefore, we plot the deviation of batch statistics to population statistics of BN in ResNet18 and Transformer$_{BN}$ in Figure 2 (top) to make a comparison. ResNet18 is trained on CIFAR-10 [17] and accuracy will drop 2 percent if we replace BN with LN. We find that at the end of the training, Transformer$_{BN}$ has a much bigger mean and variance deviation than ResNet18. Besides, the last several BN layers that are close to the output in Transformer$_{BN}$ have large variance deviation (Figure 2 (bottom)), which negatively impact the model output. Furthermore, the performance degradation of Transformer$_{BN}$ coincides with the increase of variance deviation by comparing Figure 1 (right) and Figure 2 (bottom right). Based on these observations, *we hypothesize that the inconsistency between training and inference of BN causes BN's performance degradation in neural machine translation*. We first mathematically define the training inference discrepancy of BN in the next subsection.

## 3.2 Training Inference Discrepancy

By observing Eqns. 1 and 2, the normalized output during training can be calculated as:

$$\frac{\mathbf{x}_j - \mu_{B,j}}{\sigma_{B,j}} = \left( \frac{\mathbf{x}_j - \mu_j}{\sigma_j} + \frac{\mu_j - \mu_{B,j}}{\sigma_j} \right) \frac{\sigma_j}{\sigma_{B,j}}, \;\; j = 1, 2, ..., d, \tag{4}$$

where $\sigma_{B,j} > 0$ and $\sigma_j > 0$ are the standard deviation for the $j$-th dimension. We can see $\frac{\mu_j - \mu_{B,j}}{\sigma_j}$ and $\frac{\sigma_j}{\sigma_{B,j}}$ can be viewed as random variables. Their magnitude can characterize the diversity of mini-batch examples during training and indicate how hard the estimation of population statistics is. We thus define the training inference discrepancy to quantitatively measure the inconsistency as follows.

**Definition 1** (Training Inference Discrepancy (TID)). Let $p_B$ be the distribution of batch data. Given a mini-batch data $X$ sampled from $p_B$, we define the TID of its mean and variance (with respect to model parameter $\theta$) as:

$$\text{Mean TID} = \mathbb{E}_{X \sim p_B} \frac{\|\mu_B - \mu\|_2}{\|\sigma\|_2}$$
$$\text{Variance TID} = \mathbb{E}_{X \sim p_B} \frac{\|\sigma_B - \sigma\|_2}{\|\sigma\|_2} \tag{5}$$

In terms of computing the TID in practice, we add a small positive constant in the denominator to avoid numerical instability. We save the checkpoint at the end of each epoch and before training. We first estimate the population statistics by running forward propagation one epoch and then compute mean and variance TID by another epoch.

We omit $\theta$ when it can be inferred from context without confusion. We compute the average mean and variance TID of all BN layers in ResNet18 trained on CIFAR10 and that of Transformer$_{BN}$ trained on IWSLT14 throughout training. At the end of the training, the average mean/variance TID of BN in ResNet18 is approximately 0.8%/0.9%, while that in Transformer is around 2.8%/4.1%. TID in Transformer is much larger than that in ResNet18. The trends are the same as basic observations in Section 3.1. We will use Equation (5) to compute TID in the subsequent analysis due to its better theoretical formulation (Equation (4)).

### 3.3 Comprehensive Validation

To further verify our hypothesis that large inconsistency between training and inference of BN causes BN's degraded performance, we conduct experiments on Neural Machine Translation (NMT), Language Modeling (LM), Named Entity Recognition (NER), and Text Classification (TextCls) tasks. We test both Post-Norm [39] and Pre-Norm [42] Transformers.

**Experimental Setup**  We briefly illustrate the experimental settings. More detailed description can be found in supplementary materials. For neural machine translation, we use IWSLT14 German-to-English (De-En) and WMT16 English-to-German (En-De) datasets, following the settings in Shen et al. [36]. Our code is based on *fairseq* [30][3]. For language modeling, we conduct experiments on PTB [28] and WikiText-103 (WT103) [27]. We follow the experimental settings in Shen et al. [36], Ma et al. [24]. For named entity recognition, we choose CoNLL2003 (English) [33] and Resume (Chinese) [50] datasets. We mainly follow the experimental settings in Yan et al. [44]. For text classification, we select one small scale dataset (IMDB) [25] and three large scale datasets (Yelp, DBPedia, Sogou News). We use the code[4] and follow most configurations in Bhardwaj et al. [3].

**Performance Result**  We first verify the inefficiency of BN compared to LN on four natural language tasks. Results for Post-Norm and Pre-Norm Transformers are listed in Table 1. BN performs much worse than LN on NMT, slightly worse on NER and TextCls tasks, but performs much better on LM. Although BN performs worse in most cases, it has remarkable improvement over LN on LM, raising the question: what contributes to the failure or success of BN?

**Analyzing the Statistics of BN**  We compute the TID of the last BN layer in Table 1 and leave the average TID of all BN layers in supplementary materials. The last BN layer, which is close to the output, significantly impacts the model prediction. We observe that TID is highly correlated with the performance gap between BN and LN. When TID is large, e.g., on WMT16, BN performs much worse than LN. However, when the TID of BN is negligible, e.g., on PTB and WT103, BN performs better than LN with a large margin. We select one dataset from each task with Pre-Norm Transformer and define the total TID as the sum of mean and variance TID. At the end of the training, the total TID of the last BN layer for WMT16/CoNLL/IMDB/WT103 is around 38%/16%/9%/5%, and the performance gap is -2.1 BLEU scores/-1.1 F1 score/-0.1% accuracy/6.8 perplexity (PPL). Larger TID tends to hurt BN's performance.

To explore the quantitative relation between TID and performance gap, we substitute $L = 3 \sim 6$ LN layers with BN layers from the bottom in the Post-Norm Transformer encoder on IWSLT14. As $L$ increases, the variance TID of the last BN layer grows, and the BLEU scores of Transformer$_{BN}$

---

[3]https://github.com/pytorch/fairseq. MIT license.
[4]https://github.com/declare-lab/identifiable-transformers. Apache-2.0 license.

Table 1: Results for performance and TID of last BN layer with Post-Norm (top) and Pre-Norm (bottom) Transformers on four tasks containing ten datasets. We use BLEU scores (%)/perplexity/F1 score (%)/accuracy (%) to measure the model performance on neural machine translation/language modeling/named entity recognition/text classification. "+" ("-") means the bigger (smaller) the better. Post-LN means the Post-Norm Transformer with LN. Performance gap is the difference between performance of BN and LN. Positive (Negative) Performance gap indicates BN performs better (worse) than LN.

| Task | NMT (+) | | LM (-) | | NER (+) | | | TextCls (+) | | |
|---|---|---|---|---|---|---|---|---|---|---|
| Datasets | IWSLT14 | WMT16 | PTB | WT103 | Resume | CoNLL | IMDB | Sogou | DBPedia | Yelp |
| Post-LN | 35.5 | 27.3 | 53.2 | 20.9 | 94.8 | 91.3 | 84.1 | 94.6 | 97.5 | 93.3 |
| Post-BN | 34.0 | 25.0 | 45.9 | 17.2 | 94.5 | 90.9 | 84.0 | 94.3 | 97.5 | 93.3 |
| Performance Gap | -1.5 | -2.3 | 7.3 | 3.7 | -0.3 | -0.4 | -0.1 | -0.3 | 0 | 0 |
| Mean TID of $BN_{last}$ | 1.5% | 4.2% | 0.9% | 1.8% | 1.7% | 4.2% | 1.8% | 1.8% | 2.2% | 3.1% |
| Var TID of $BN_{last}$ | 10.6% | 17.9% | 1.1% | 2.0% | 3.7% | 9.5% | 3.9% | 4.3% | 3.5% | 4.0% |
| Pre-LN | 35.5 | 27.3 | 54.5 | 24.6 | 94.0 | 91.0 | 84.1 | 94.5 | 97.5 | 93.3 |
| Pre-BN | 34.8 | 25.2 | 45.9 | 17.8 | 93.2 | 89.9 | 84.0 | 94.3 | 97.5 | 93.3 |
| Performance Gap | -0.7 | -2.1 | 8.6 | 6.8 | -0.8 | -1.1 | -0.1 | -0.2 | 0 | 0 |
| Mean TID of $BN_{last}$ | 3.4% | 7.9% | 1.6% | 2.4% | 9.6% | 10.0% | 2.9% | 7.5% | 3.9% | 12.1% |
| Var TID of $BN_{last}$ | 4.6% | 30.1% | 1.7% | 2.5% | 6.5% | 6.4% | 6.2% | 7.1% | 3.3% | 8.6% |

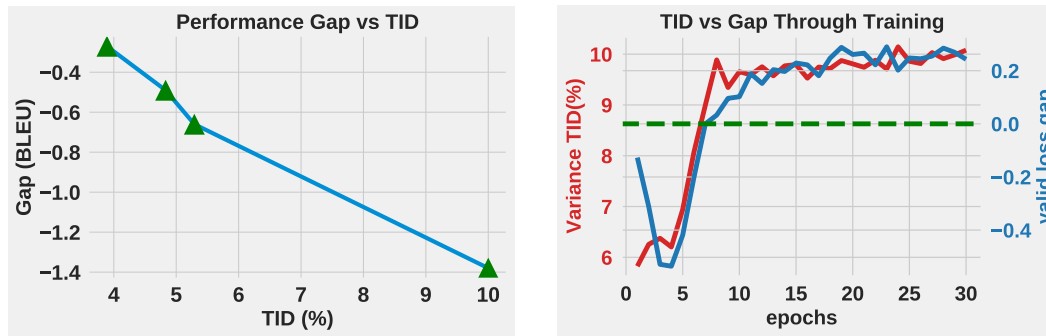

Figure 3: Left: Variance TID and BLEU gap between Transformer$_{BN}$ and Transformer$_{LN}$ when replacing different numbers of LN layers with BN. Right: Variance TID and valid loss gap of Post-Norm Transformer through training.

drops off. We plot the variance TID and BLEU gap between Transformer$_{BN}$ and Transformer$_{LN}$ in Figure 3 (left). We can see that the two quantities are highly correlated.

In Figure 3 (right), we plot the variance TID of the last BN layer and the validation loss gap between Transformer$_{BN}$ and Transformer$_{LN}$ on IWSLT14 through training. The validation loss gap is calculated by subtracting loss of Transformer$_{LN}$ from Transformer$_{BN}$. At the beginning of training, BN performs better than LN. When the TID begins to explode, BN's performance starts to degrade.

Based on the results in Table 1 and observations in Figure 3, we argue that TID serves as an indicator of BN's performance in Transformers. Large TID hurts BN's performance, while BN with small TID performs better than LN due to its more efficient optimization (see experimental validation in Section 4.3).

## 4 Suppressing High TID by RBN

In this section, we are devoted to reducing the TID of BN when it is large. If TID is suppressed, the performance of BN will be improved and may exceed LN due to the training efficiency of BN.

### 4.1 Regularized Batch Normalization

Assume there are $H$ layers of BN in a neural network. We denote the batch statistics and running statistics of each layer by $\mu_B^i$, $\sigma_B^i$, and $\mu^i$, $\sigma^i$, $i = 1 \ldots, H$. Assume the Cross-Entropy (CE) loss with respect to the neural network parameters $\theta$ is denoted by $\mathcal{L}(\theta)$. To avoid undesirable training

Table 2: Results for the performance of Post-Norm (top) and Pre-Norm (bottom) Transformers with LN/BN/RBN. RBN consistently improves BN and could match or exceed LN on 17 out of 20 settings.

| Task | NMT (+) | | LM (-) | | NER (+) | | TextCls (+) | | | |
|------|---------|--|--------|--|---------|--|-------------|--|--|--|
| Datasets | IWSLT14 | WMT16 | PTB | WT103 | Resume | CoNLL | IMDB | Sogou | DBPedia | Yelp |
| Post-LN | **35.5** | **27.3** | 53.2 | 20.9 | **94.8** | 91.3 | 84.1 | 94.6 | 97.5 | 93.3 |
| Post-BN | 34.0 | 25.0 | 45.9 | 17.2 | 94.5 | 90.9 | 84.0 | 94.3 | 97.5 | 93.3 |
| Post-RBN | **35.5** | 26.5 | **44.6** | **17.1** | **94.8** | **91.4** | **84.5** | **94.7** | **97.6** | **93.6** |
| Pre-LN | 35.5 | **27.3** | 54.5 | 24.6 | **94.0** | **91.0** | 84.1 | 94.5 | **97.5** | 93.3 |
| Pre-BN | 34.8 | 25.2 | 45.9 | 17.8 | 93.2 | 89.9 | 84.0 | 94.3 | **97.5** | 93.3 |
| Pre-RBN | **35.6** | 26.2 | **43.2** | **17.1** | **94.0** | 90.6 | **84.4** | **94.7** | **97.5** | **93.5** |

inference discrepancy, we pose the optimization as a constrained problem:

$$\min_{\theta} \quad \mathcal{L}(\theta)$$
$$s.t. \quad \mathbb{E}_{p_B} d_\mu(\mu_B^i, \mu^i) \le \epsilon_i, \ i = 1, \ldots, H \qquad (6)$$
$$\mathbb{E}_{p_B} d_\sigma(\sigma_B^i, \sigma^i) \le \eta_i, \ i = 1, \ldots, H$$

where $d_\mu$ and $d_\sigma$ measure the inconsistency of mean and variance. This is equivalent to

$$\min_{\theta} \quad \mathcal{L}(\theta) + \sum_{i=1}^{H} \lambda_i \mathbb{E} d_\mu(\mu_B^i, \mu^i) + \nu_i \mathbb{E} d_\sigma(\sigma_B^i, \sigma^i) \qquad (7)$$

To simplify the problem, we set $\lambda_i = \lambda$, $\nu_i = \nu$, for $i = 1, \ldots, H$.

When handling batch data, we apply gradient-based optimization to the following loss ($\mathcal{L}_B(\theta)$ is the batch CE loss):

$$\mathcal{L}_B(\theta) + \sum_{i=1}^{H} \lambda d_\mu(\mu_B^i, \mu^i) + \nu d_\sigma(\sigma_B^i, \sigma^i)$$

In particular, we choose $d_\mu(\mu_B, \mu) = \|\mu_B - \mu\|_2^2$ and $d_\sigma(\sigma_B, \sigma) = \|\sigma_B - \sigma\|_2^2$. The sensitivity analysis of hyperparameter is given in Section 4.3. Since back propagating through the running statistics $\mu$ and $\sigma$ would trace back to the first batch of data which is impractical, we simply stop the gradient of $\mu$ and $\sigma$ in back propagation.

## 4.2 Experimental Result for RBN

We choose $\lambda, \nu$ both from $\{0, 0.01, 0.1, 1\}$ by validation loss. Results are shown in Table 2. The optimal hyperparameters are listed in supplementary materials.

**Neural Machine Translation** On IWSLT14 datasets, we see that RBN significantly improves BN and can exceed LN with 0.1 BLEU scores with Pre-Norm Transformer and match LN with Post-Norm Transformer. On WMT16 dataset, although RBN still falls behind LN, it could improve 1.5/1.0 BLEU scores over BN in Post-Norm/Pre-Norm setting. The reason is that even though RBN can suppress a large amount of TID, the remaining is still large since the original TID is huge. We speculate that the high data diversity in WMT16 contributes to the explosive TID of BN, which is hard to remove. We leave the verification as future work.

**Language Modeling** On Post-Norm Transformer, BN could boost the testing PPL of LN from 53.2 to 45.9 on PTB and from 20.9 to 17.2 on WikiText-103. Furthermore, substituting RBN for BN improves the testing PPL to 44.6 on PTB and 17.1 on WikiText-103. On Pre-Norm Transformer, BN elevates the testing PPL of LN from 54.5 to 45.9 on PTB and from 24.6 to 17.8 on WikiText-103. Moreover, replacing BN with RBN improves the testing PPL to 43.2 on PTB and 17.1 on WikiText-103. Overall, RBN exceeds LN with 8.6/3.8 testing PPL with Post-Norm Transformer and 11.3/7.5 testing PPL with Pre-Norm Transformer on PTB/WikiText-103.

**Named Entity Recognition** BN performs worse than LN on both Resume and CoNLL2003 datasets, especially for Pre-Norm Transformer. RBN improves BN in all settings, matches or exceeds LN in three out of four settings. By taking the better performance of Post-Norm and Pre-Norm, RBN matches the performance of LN on Resume and exceeds LN on CoNLL2003.

Table 3: Results for the performance of Post-Norm (top) and Pre-Norm (bottom) Transformers with PN/BRN/MABN/RBN.

| Task | NMT (+) | | LM (-) | | NER (+) | | | TextCls (+) | | |
|---|---|---|---|---|---|---|---|---|---|---|
| Datasets | IWSLT14 | WMT16 | PTB | WT103 | Resume | CoNLL | IMDB | Sogou | DBPedia | Yelp |
| Post-PN-only | 0 | 0 | 254.6 | inf | 94.4 | 67.1 | 84.2 | 90.6 | 97.1 | 89.6 |
| Post-PN+LS | **35.6** | 0 | 49.8 | 21.0 | 94.3 | 90.9 | 84.0 | 94.6 | 97.4 | 93.2 |
| Post-BRN | 35.3 | 25.8 | 45.1 | 17.3 | 93.6 | 89.9 | 83.6 | 94.5 | 97.5 | 93.3 |
| Post-MABN | 0 | 0 | 47.4 | 33.6 | 94.4 | 90.8 | 84.1 | 94.5 | 97.5 | 93.5 |
| Post-RBN | 35.5 | **26.5** | **44.6** | **17.1** | **94.8** | **91.4** | **84.5** | **94.7** | **97.6** | **93.6** |
| Pre-PN-only | 34.5 | 26.0 | 48.6 | inf | 5.0 | 11.1 | 84.2 | 94.4 | 97.4 | 93.3 |
| Pre-PN+LS | **35.6** | **27.2** | 59.8 | 20.9 | 93.3 | 54.1 | 83.3 | 94.4 | 97.3 | 93.4 |
| Pre-BRN | 35.2 | 25.3 | 45.7 | 17.5 | 94.1 | **91.1** | 84.3 | 94.5 | 97.4 | 93.4 |
| Pre-MABN | 35.0 | 26.8 | 48.7 | inf | **94.8** | 90.9 | **84.4** | 94.6 | 97.5 | 93.3 |
| Pre-RBN | **35.6** | 26.2 | **43.2** | **17.1** | 94.0 | 90.6 | **84.4** | **94.7** | 97.5 | 93.5 |

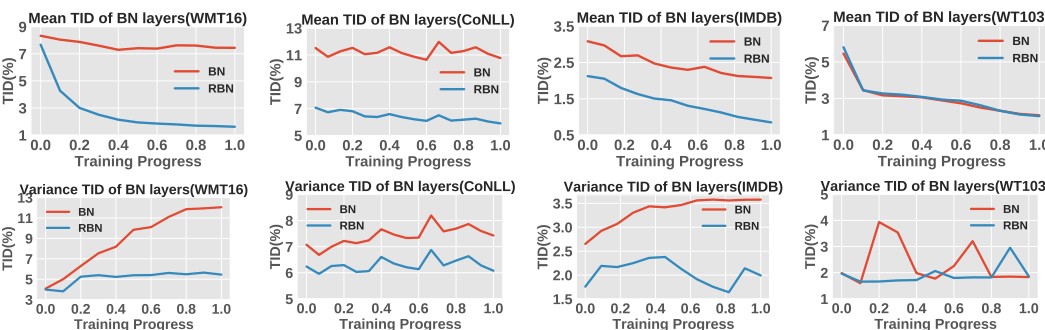

Figure 4: Average Mean and Variance TID on WMT16/CoNLL/IMDB/WT103 for Pre-Norm Transformer with BN and RBN. RBN reduces the Mean and Variance TID of BN at the end of the training and leads to better performance.

**Text Classification**  We find that BN performs similar to/worse than LN on 4/4 settings. RBN improves the performance of BN consistently and can match/exceed LN on 1/7 settings. RBN improves BN with 0.3% accuracy on average, which shows the benefit of our regularization. We do not intend to achieve the state-of-the-art performance but to verify the efficacy of RBN.

**Comparison to BN's Variants**  We compare our RBN with Power Normalization (PN) [36], Batch Renormalization (BRN) [14], and Moving Averaing Batch Normaliazation (MABN) [45] in Table 3. These methods incorporate population statistics of BN in training, which is beneficial for alleviating training inference inconsistency of BN. PN and MABN are implemented by their official codes[5]. BRN is implemented according to their paper [14]. The configurations of PN, BRN, and MABN are given in supplementary materials. We highlight that PN incorporates layer scaling (LS) [49], which is important for stabilizing training, as shown in the supplementary materials and official code of PN. We thus report the results for PN only and PN with layer scaling (PN+LS). We can see that RBN performs the best in most settings. PN and MABN is not stable without layer scaling, especially for Post-Norm Transformers.

### 4.3  Analysis

**Training Inference Inconsistency**  We compute the TID of the last BN layer ($BN_{last}$) in Table 4 and plot the average TID of BN and RBN on WMT16, WT103, CoNLL2003, and IMDB datasets for Pre-Norm Transformers through training in Figure 4. Figures of TID for other datasets and Post-Norm Transformer can be found in supplementary materials. We can see that RBN reduces BN's mean and variance TID at the end of training. On neural machine translation and named entity recognition tasks, the original TID is large. RBN significantly decreases the TID of

---

[5]https://github.com/sIncerass/powernorm. GPL-3.0 license. https://github.com/megvii-model/MABN. MIT license.

Table 4: TID of the last BN/RBN layer in Post-Norm and Pre-Norm Transformers on various NLP tasks. RBN reduces the TID of BN effectively.

| Task | NMT | | LM | | NER | | TextCls | | | |
|---|---|---|---|---|---|---|---|---|---|---|
| Datasets | IWSLT14 | WMT16 | PTB | WT103 | Resume | CoNLL | IMDB | Sogou | DBPedia | Yelp |
| Post-Norm Transformer | | | | | | | | | | |
| Mean TID of $BN_{last}$ | 1.5% | 4.2% | 0.9% | 1.8% | 1.7% | 4.2% | 1.8% | 1.8% | 2.2% | 3.1% |
| Mean TID of $RBN_{last}$ | 0.8% | 2.3% | 0.9% | 1.8% | 1.4% | 1.9% | 0.2% | 0.2% | 0.3% | 0.2% |
| Var TID of $BN_{last}$ | 10.6% | 17.9% | 1.1% | 2.0% | 3.7% | 9.5% | 3.9% | 4.3% | 3.5% | 4.0% |
| Var TID of $RBN_{last}$ | 6.7% | 7.7% | 1.1% | 1.7% | 3.0% | 5.0% | 1.2% | 0.2% | 0.3% | 0.1% |
| Pre-Norm Transformer | | | | | | | | | | |
| Mean TID of $BN_{last}$ | 3.4% | 7.9% | 1.6% | 2.4% | 9.6% | 10.0% | 2.9% | 7.5% | 3.9% | 12.1% |
| Mean TID of $RBN_{last}$ | 3.2% | 1.3% | 1.6% | 2.4% | 4.5% | 4.0% | 0.7% | 1.0% | 1.1% | 1.0% |
| Var TID of $BN_{last}$ | 4.6% | 30.1% | 1.7% | 2.5% | 6.5% | 6.4% | 6.2% | 7.1% | 3.3% | 8.6% |
| Var TID of $RBN_{last}$ | 1.5% | 12.1% | 1.7% | 2.4% | 6.3% | 5.6% | 4.7% | 0.4% | 0.5% | 0.5% |

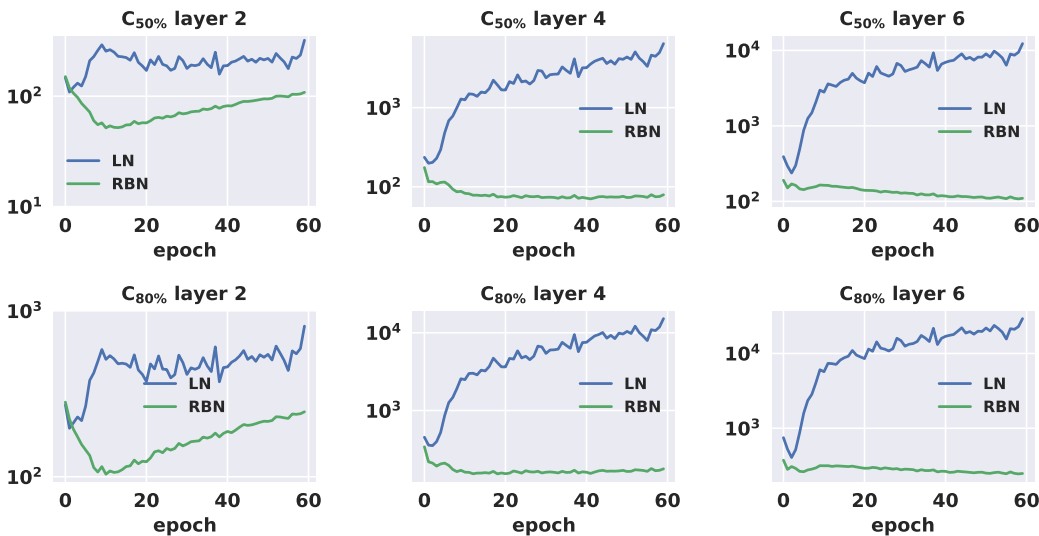

Figure 6: $C_{50\%}$ (top), and $C_{80\%}$ (bottom) of input features of Transformer encoder layer 2/4/6. RBN improves the $C_{50\%}$ and $C_{80\%}$ of LN, especially for deep layers (2 orders of magnitude at layer 6).

BN and improves BN's performance by a clear margin. For language modeling and text classification tasks, RBN also reduces the moderate TID of BN and gets better PPL or accuracy.

**Sensitivity to Hyperparameters** We test different penalty coefficients for RBN on neural machine translation with Pre-Norm Transformer. The results are shown in Figure 5. Penalizing the mean and variance discrepancy can both improve the performance of BN. Combining them with moderate coefficients achieves the best performance.

**Training Dynamics** To show the optimization advantages of RBN over LN, we explore the layer-wise training dynamics of LN and RBN in Pre-Norm Transformer on IWSLT14. We refer the reader to Huang et al. [12] for detailed analysis about the correlation between optimization of neural network and layer-wise training dynamics. We empirically observe that replacing LN with RBN significantly improves the layer-wise conditioning [12] of Transformer. We denote the intermediate embedding in Transformer by $\tilde{\mathbf{X}} \in \mathbb{R}^{B \times T \times d}$, each $\tilde{\mathbf{X}}_{i,j,:} \in \mathbb{R}^d$ is a

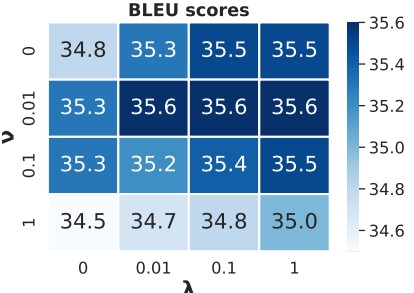

Figure 5: The BLEU scores on IWSLT14 with different mean ($\lambda$) and variance ($\nu$) discrepancy penalty of RBN.

word vector. We reshape $\tilde{\mathbf{X}}$ to a sequence of word vectors to $\mathbf{X} = [\mathbf{x}_1, \mathbf{x}_2, \ldots, \mathbf{x}_{BT}] \in \mathbb{R}^{BT \times d}$. We assume $BT > d$ which is satisfied in our experiments. We define the general condition number with

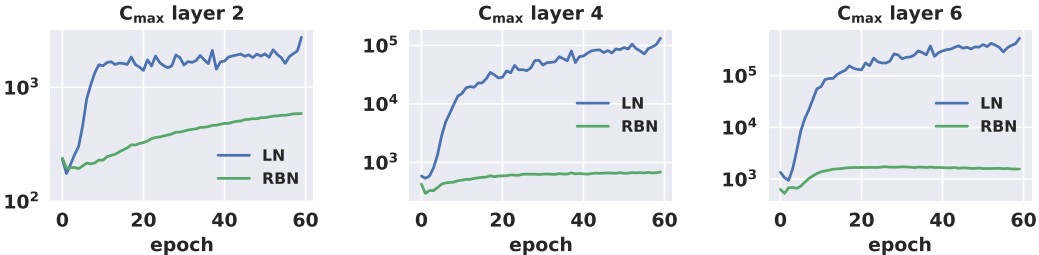

Figure 7: $C_{max}$ of input features of Transformer encoder layer 2/4/6 through training.

respect to the percentage as $C_p(\mathbf{X}) = \frac{\sigma_1}{\sigma_{\lceil pd \rceil}}, 0 < p \leq 1$. $\lceil a \rceil$ is the smallest integer that is larger than or equal to $a$. Lower $C_p(\mathbf{X})$ is usually associated with faster convergence of training. We plot $C_{50\%}$, and $C_{80\%}$ of input features of transformer encoder layer 2/4/6 in Figure 6. We can see that RBN significantly reduces the $C_{50\%}$ and $C_{80\%}$ of LN, usually with orders of magnitude. We also plot the layer-wise $C_{max}(\mathbf{X}) = \lambda_{max}((\mathbf{X}^T\mathbf{X})^{\frac{1}{2}})$ in Figure 7. Smaller $C_{max}$ usually permits higher learning rates which leads to faster training and better generalization [10]. RBN has much smaller $C_{max}$ than LN.

## 5    Conclusion and Limitation

In this paper, we defined Training Inference Discrepancy (TID) and showed that TID is a good indicator of BN's performance for Transformers, supported by comprehensive experiments. We observed BN performs much better than LN when TID is negligible and proposed Regularized BN (RBN) to alleviate TID when TID is large. Our RBN has theoretical advantages in optimization and works empirically better by controlling the TID of BN when compared with LN. We hope our work will facilitate a better understanding and application of BN in NLP.

**Limitation.**   Our analyses on TID are almost empirical studies without theoretical guarantee. It is better to further model the geometric distribution of word embedding, evolving along with the training dynamics and information propagation, with theoretical derivation under mild assumptions. Besides, our proposed RBN cannot entirely suppress huge TID in training large-scale datasets with high diversity, leading to degraded performance. One possible direction is to combine RBN and LN for both better optimization properties and small TID, as explored in [13, 46] for CV tasks.

**Acknowledgement**   Ji Wu was sponsored by National Key Research & Development Program of China (2021ZD0113402), Tsinghua University Spring Breeze Fund (2021Z99CFZ010), National Key Research & Development Program of China (Grant Number: 2021YFC2500803), and Tsinghua-Toyota Joint Research Institute Inter-disciplinary Program. Lei Huang was supported by National Natural Science Foundation of China (Grant No. 62106012) and the Fundamental Research Funds for the Central Universities.

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
