# OpenReview forum: "Understanding the Failure of Batch Normalization for Transformers in NLP"
_NeurIPS.cc/2022/Conference — NeurIPS 2022 Accept_

### Official Review · Reviewer_fs4E · 2022-06-28

**Rating:** 5
**Confidence:** 3
**Soundness:** 3 good
**Presentation:** 3 good
**Contribution:** 3 good

**Summary:**

The authors begin by relating the failure of BN variance in NLP models. They define Training Inference Discrepancy (TID) to quantitatively measure this inconsistency and show that TID can serve as an indicator of BN’s performance.
In particular, BN reaches much better test performance than LN when TID keeps small through training (e.g., in the CV case).
They propose Regularized BN (RBN) that adds a regularization term in BN to penalize and reduce the TID when the TID of BN is large.
In experimetns, RBN can outperform BN in a lot of settings, and outperform LN in some settings.

**Questions:**

I  don't have any question at this point.

**Limitations:**

Yes.

**Strengths And Weaknesses:**

Strength:
The proposed approach is novel and the point of view is convincing.
It's good to address the BN mystery in NLP.

Weakness:
No theory advancement.
Empirical improvements are not large.

---

> ### Author Response · Authors · 2022-08-02
> **Response to Reviewer fs4E**
>
> We thank the reviewer for the encouraging and insightful comments.

---

### Official Review · Reviewer_syax · 2022-07-11

**Rating:** 6
**Confidence:** 3
**Soundness:** 3 good
**Presentation:** 3 good
**Contribution:** 3 good

**Summary:**

This paper attempts to explain the failure of Batch Normalization (BN) in NLP tasks. The authors find that the inconsistency between training and inference leads to the failure. They define Training Inference Discrepancy (TID) to quantitatively measure this inconsistency and show that BN can obtain better performance when TID keeps small during training. Based on this observation, the authors propose Regularized BN that adds a regularization term of TID during training. Experiences show that RBN can improve the performance of BN and outperforms or is on par with layer normalization. The authors also conduct some analyses to show the advantages of their method.

**Questions:**

This paper can be further improved if the author could provide more explanation about their observation. Here are some questions that I hope the authors can answer:
1.	Can you explain why the performance of BN on language modeling tasks is already better than LN?
2.	Can you explain why Transformer_BN has a much bigger mean and variance deviation than ResNet18?
3.	Can you compare your RBN with Powernorm?


**Limitations:**

The authors mention that they don’t have a theoretical guarantee about their method in the Limitation section. I agree with it. But I think this work contains sufficient empirical findings that can benefit the community.

There is no potential negative social impact of this work.


**Strengths And Weaknesses:**

Strengths
1.	This paper focuses on a well-known learning problem in NLP field.
2.	This paper finds an interesting observation from extensive experiments across different CV and NLP tasks.
3.	Based on their observation, the authors proposed a simple yet effective method.
4.	This paper is well-written and easy to follow

Weaknesses
1.	Lack of theoretical explanation and guarantee.
2.	Without compared with other related normalization methods, for example, Powernorm.



Novelty:
This paper is novel. The failure of Batch Normalization in NLP model is a well-known learning problem in NLP field. This paper is the first one to find that the inconsistency between training and inference and the performance gap between batch normalization and layer normalization are highly correlated. Based on this observation, the authors proposed a novel and simple method that can consistently improve the performance of batch normalization.

Soundness:
The proposed method is empirically validated on different tasks and dataset. But there is no theocratical guarantee about the soundness of their method.

Significance:
The proposed RBN method can consistently improve the performance of Batch Normalization to on be par with Layer Normalization on different tasks and dataset.
However, the author didn’t compare their method with other batch normalization methods, for example, Powernorm. They only discuss the difference in Section2.

Presentation:
This paper is well-organized. Starting from an observation, the authors raise a hypothesis and conduct extensive experiments to validate their hypothesis. Based on that, the authors propose a method that can significantly improve the performance. Finally, the authors conduct analyses on the model trained with their method to validate their understanding.

---

> ### Author Response · Authors · 2022-08-02
> **Response to Reviewer syax**
>
> We thank the reviewer for the encouraging and insightful comments. Please find our responses to specific questions below.
>
> **Question 1**: Can you explain why the performance of BN on language modeling tasks is already better than LN?
>
> **Response:**  Language modeling is essentially an classification task with large number of  classes (number of words in vocabulary). The data, especially wikitext 103 dataset,  is relatively hard to be fitted. Thus, optimization plays a more important role in language modeling task. BN offers better optimization performance and properties compared to LN, but has a lack of training inference consistency. With our quantitative definition of training inference inconsistency, we find that BN has very small inconsistency on language modeling tasks.  Thus, BN exert its advantage on optimization and performs better than LN on language modeling tasks.
>
>
>
> **Question 2:** Can you explain why Transformer_BN has a much bigger mean and variance deviation than ResNet18?
>
> **Response:**  We think both data and architecture contribute to the large deviation. If we use BN in Vision Transformer (ViT-BN), the mean and variance deviation of ViT-BN fall between Transformer_BN and ResNet18.
>
>
> **Question 3:** Can you compare your RBN with Powernorm?
>
> **Response:**  We compare the performance of RBN with Batch Renormalization (BRN) [14], Moving Averaing Batch Normaliazation (MABN) [44], and PowerNorm [36].
> BRN and MABN are helpful to decrease training inference inconsistency in small batch setting in CV tasks.  We highlight that PowerNorm incorporates a scaling layer (the root mean square layer normalization mehtod [48]), which is important for stabilizing training, as shown in the supplementary materials and official code of PowerNorm. We thus also compare PowerNorm-only and PowerNorm+layerscaing.
> From the results shown in following table, we can see that RBN performs the best in most settings. PN is not stable without layer scaling.
>
> |        norm\dataset        | IWSLT14  |  WMT16   |   PTB    |  WT103   |
> | :------------------------: | :------: | :------: | :------: | :------: |
> |          Post-RBN          |   35.5   | **26.5** | **44.6** | **17.1** |
> |    Post-PowerNorm-only     |    0     |    0     |  254.6   |   inf    |
> | Post-PowerNorm+layerscaing | **35.6** |    0     |   49.8   |   21.0   |
> |          Post-BRN          |   35.3   |   24.8   |   45.1   |   17.3   |
> |         Post-MABN          |    0     |    0     |   47.4   |   33.6   |
> |                    |  |     |  |  |
> |          Pre-RBN           | **35.6** |   26.2   | **43.2** | **17.1** |
> |     Pre-PowerNorm-only     |   34.5   |   26.0   |   48.6   |   inf    |
> | Pre-PowerNorm+layerscaing  | **35.6** | **27.2** |   59.8   |   20.9   |
> |          Pre-BRN           |   35.2   |   25.3   |   45.7   |   17.5   |
> |          Pre-MABN          |   35.0   |   25.8   |   48.7   |   inf    |
>
> |        norm\dataset        |  Resume  |  CoNLL   |   IMDB   |  Sogou   | DBPedia  |   Yelp   |
> | :------------------------: | :------: | :------: | :------: | :------: | :------: | :------: |
> |          Post-RBN          | **94.8** | **91.4** | **84.5** | **94.7** | **97.6** | **93.6** |
> |    Post-PowerNorm-only     |   94.4   |   67.1   |   84.2   |   90.6   |   97.1   |   89.6   |
> | Post-PowerNorm+layerscaing |   94.3   |   90.9   |   84.0   |   94.6   |   97.4   |   93.2   |
> |          Post-BRN          |   93.6   |   89.9   |   83.6   |   94.5   |   97.5   |   93.3   |
> |         Post-MABN          |   94.4   |   90.8   |   84.1   |   94.5   |   97.5   |   93.5   |
> |            |     |      |  |  |  |  |
> |          Pre-RBN           |   94.0   |   90.6   | **84.4** | **94.7** | **97.5** | **93.5** |
> |     Pre-PowerNorm-only     |   5.0    |   11.1   |   84.2   |   94.4   |   97.4   |   93.3   |
> | Pre-PowerNorm+layerscaing  |   93.3   |   54.1   |   83.3   |   94.4   |   97.3   |   93.4   |
> |          Pre-BRN           |   94.1   | **91.1** |   84.3   |   94.5   |   97.4   |   93.4   |
> |          Pre-MABN          | **94.8** |   90.9   | **84.4** |   94.6   | **97.5** |   93.3   |
>
> We only run one seed for PowerNorm, BRN, and MABN.
> For PowerNorm, we follow their source code. We use 4000 warmups, and set foward and backward momentum as $0.9$.
> For BRN, we use one epoch BN as  warmup and linearly increase $r$ to $3$ and $d$ to $5$, which are the same as BRN paper. $r$ and $d$ are renormalizing factors.
> For MABN,  we use $16$ mini-batches to compute simple moving average statistics and momentum $\alpha=0.98$ to compute exponential moving average statistics,  which are the same as MABN paper.
>
> Thank you for suggesting the comparison to other methods. We will include the results in our final manuscript.

---

> > ### Comment · Reviewer_syax · 2022-08-09
> > **Thanks for the responses.**
> >
> > The additional experimental results should be included in the revised version to make this work more convincing.

---

> > > ### Author Response · Authors · 2022-08-09
> > > **Response to Reviewer syax**
> > >
> > > Sure, we will add the additional results to the revised version. Thanks!

---

### Official Review · Reviewer_vog4 · 2022-07-13

**Rating:** 6
**Confidence:** 3
**Soundness:** 3 good
**Presentation:** 3 good
**Contribution:** 3 good

**Summary:**

This paper explores potential explanations for the lack of effectiveness of batch norm compared to LayerNorm. They find a phenomenon that correlates with this (TID) and propose a method to tackle this (RBN).

---

After response: Thank you for your response, I appreciate the additional clarifications and experiments!

---

**Questions:**

### Questions:
- Does this finding invalidate the PowerNorm findings? Or is simply an added observation.
- TID seems to be clearly correlated with performance, however do you have any intuition on why this discrepancy occurs?
- Furthermore, why is this discrepancy smaller on some tasks versus others? Is this something to do with distributional shift on the dataset or the nature of the task?

### Missing Citations:
[Understanding and Improving Layer Normalization](https://arxiv.org/pdf/1911.07013)

**Limitations:**

They address the main limitation in their paper, I appreciate this.

**Strengths And Weaknesses:**

Pros:
- Solid finding about the correlation between TID and BN
- Good presentation

Cons:
- Not much discussion on **why** this happens...which I feel would be a greater contribution for future work to build upon.
- Furthermore, it is unclear whether RBN is superior to LayerNorm. One claim related to its use in practice is that RBN tends to converge faster, however this is not supported directly. It would be nice to have

---

> ### Author Response · Authors · 2022-08-02
> **Response to Reviewer vog4**
>
> We thank the reviewer for the encouraging and insightful comments. Please find our responses to specific questions and concerns below.
>
> **Question 1:** Does this finding invalidate the PowerNorm findings? Or is simply an added observation.
>
> **Response:** PowerNorm observes that the forward and backward statistics of BN are more diverse (unstable) in NLP than that in CV. We further find that the diversity of statistics of BN varies across different NLP tasks.
> Our definition of training inference discrepancy is scale invariant, while the deviation defined in PowerNorm depends on the scale of input. PowerNorm defines the mean and variance deviation as $\frac{1}{d}\Vert  \mu -\mu_{B}\Vert$ and $\frac{1}{d}\Vert  \sigma^2 -\sigma^2_{B}\Vert$ in Figure 2 of the PowerNorm paper.  BN is  typically added after linear transformation, i.e., $y=BN(Wx)$. The mean and variance deviation defined in PowerNorm will increase by $k$ and $k^2$ if we multiply the weight $W$ or feature $x$ by a factor of $k$. Furthermore, we find our defined TID can well serve as an performance indicator of BN. Even though our finding does not invalidate the PowerNorm finding, our work moves a non-trivial step beyond their finding.
>
>
>
> **Question 2:** TID seems to be clearly correlated with performance, however do you have any intuition on why this discrepancy occurs?
>
> **Response:** The batch statistics could be far from the population statistics and the TID will be large, when the empirical distribution of batch data is distant from the population distribution.  In NLP, we think the word distribution of sentences can vary significantly. For instance, different sentences come from different topics, each topic has its own word distribution. Large sentence variation makes it hard for batch distribution to approximate population distribution, leading to large TID. We also conjecture that the Transformer architecture contributes to large TID of BN since LN, rather than BN, is widely used in various Vision Transformers for CV tasks. We leave this exploration as future work.
>
>
>
> **Question 3:** Furthermore, why is this discrepancy smaller on some tasks versus others? Is this something to do with distributional shift on the dataset or the nature of the task?
>
> **Response:** This discrepancy (TID) is defined on training set and is irrelevant to testing distribution, thus it is not caused by the distributional shift between training and testing. We think the distributional shift is an impact factor that is orthogonal to TID. Testing BN with batch statistics could mitigate the distributional shift to some extent. We show the results in supplementary material (part E, table 4). Testing BN with batch statistics can improve BN in certain settings, but still perform worse than LN and RBN. We think different discrepancies stem from the nature of tasks, e.g., data diversity, network architectures.
>
>
>
> **Concern 1:** It is unclear whether RBN is superior to LayerNorm. One claim related to its use in practice is that RBN tends to converge faster, however this is not supported directly. It would be nice to have.
>
> **Response:** One advantage of RBN over LayerNorm is that RBN inherits the merit of BN in optimization. The following table shows the training nll loss of Post-Norm Transformer RBN and Transformer LN on IWSLT14 (top) and WMT16 (bottom). We can find that the training of RBN is faster than LN. Another advantage of RBN is that it does not introduce additional computation cost during inference (we can merge the population statistics into the linear transformation after training), while LN has to perform normalization again during inference for each data.
>
> | norm\epoch |  5   |  10  |  30  |  50  |
> | :--------: | :--: | :--: | :--: | :--: |
> |     LN     | 5.33 | 2.97 | 2.67 | 2.38 |
> |    RBN     | 4.77 | 2.95 | 2.65 | 2.37 |
>
>
> | norm\epoch |  5   |  10  |  15  |  20  |
> | :--------: | :--: | :--: | :--: | :--: |
> |     LN     | 3.41 | 3.04 | 2.87 | 2.83 |
> |    RBN     | 3.36 | 3.00 | 2.84 | 2.80 |
>
>
>
> **Missing citation:** We thank the reviewer and will add the missing citation in the revised manuscript.

---

### Meta-Review · Area_Chair_ZoS2 · 2022-08-26

**Recommendation:** Accept
**Confidence:** Less certain

**Metareview:**

The paper studies the reason why Batch Normalization is not effective in NLP tasks. The authors find that the inconsistency between training and inference leads to the failure. They define Training Inference Discrepancy (TID) to measure the inconsistency and show that BN can obtain better performance when TID is small. The authors propose Regularized BN with an additional regularization term. Experiments show RBN is better than plain BN and comparable to Layer Normalization.
Authors may want incorporate the additional analysis in the feedback.

**Award:**

No

---

### Decision · Program_Chairs · 2022-09-14

Accept